# Visual Classification via Description from Large Language Models

**Sachit Menon, Carl Vondrick**
Department of Computer Science
Columbia University

## Abstract

Vision-language models (VLMs) such as CLIP have shown promising performance on a variety of recognition tasks using the standard zero-shot classification procedure – computing similarity between the query image and the embedded words for each category. By only using the category name, they neglect to make use of the rich context of additional information that language affords. The procedure gives no intermediate understanding of why a category is chosen, and furthermore provides no mechanism for adjusting the criteria used towards this decision. We present an alternative framework for classification with VLMs, which we call classification by description. We ask VLMs to check for descriptive features rather than broad categories: to find a tiger, look for its stripes; its claws; and more. By basing decisions on these descriptors, we can provide additional cues that encourage using the features we want to be used. In the process, we can get a clear idea of what features the model uses to construct its decision; it gains some level of inherent explainability. We query large language models (e.g., GPT-3) for these descriptors to obtain them in a scalable way. Extensive experiments show our framework has numerous advantages past interpretability. We show improvements in accuracy on ImageNet across distribution shifts; demonstrate the ability to adapt VLMs to recognize concepts unseen during training; and illustrate how descriptors can be edited to effectively mitigate bias compared to the baseline.

## 1 Introduction

Why does a person recognize a hen in Fig.1? If you had to justify your answer, you might name its beak, describe its feathers, or discuss any number of other traits that we associate with hens. It is easy for people to describe the visual features of categories in words, as well as use these verbal descriptions to aid perception. However, generating such schemata, let alone leveraging them for perceptual tasks, has remained a key challenge in machine learning.

Vision-language models (VLMs) trained on large corpora of paired image-text data, such as CLIP (Radford et al., 2021), have seen huge successes recently, dominating image classification. The standard zero-shot classification procedure – computing similarity between the query image and the embedded words for each category, then choosing the highest – has shown impressive performance on many popular benchmarks, such as ImageNet (Russakovsky et al., 2015). Comparing to the word

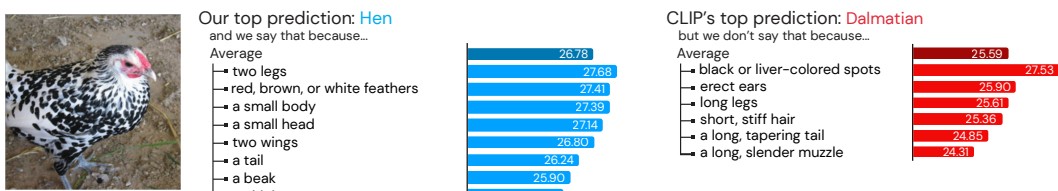

Figure 1: On the left, we show an example decision by our model in addition to its justification (blue bars). On the right, we show how CLIP classifies this image. Our model does not make the same mistake because it cannot produce a compatible justification with the image (red bars).

that names a category was a reasonable place to start because these methods can rely on the fact that the word "hen" tends to show up near images of hens on the Internet.

Despite the advances on classification performance, the large models often make unreasonable mistakes or give undesired answers (Goh et al., 2021). The standard zero-shot method gives us no intermediate understanding (i.e. explanation) of the model's reasoning process. They often fail to look at cues that a human would use easily, and there is no clear way to get the right cues or provide them to the model.

Our key insight is that we can use language as an internal representation for visual recognition, which creates an interpretable bottleneck for computer vision tasks. Instead of querying a VLM with just a category name, the use of language enables us to flexibly compare to *any* words. If we have an idea what features should be used, we can ask the VLM to check for those features instead of just the class name. To find a hen, look for its beak; its feathers; and more. By basing the decision on these features, we can provide additional cues that encourage looking at the features we want to be used. In the process, we can get a clear idea of what the model uses to make its decision; it is inherently explainable.

However, hand-writing these features can be costly, and does not scale to large numbers of classes. We can solve this by requesting help from another model. Large language models (LLMs), such as GPT-3 (Brown et al., 2020), show remarkable world knowledge on a variety of topics. They can be thought of as implicit knowledge bases, noisily condensing the collective knowledge of the Internet in a way that can be easily queried with natural language (Petroni et al., 2019). As people often write about what things look like, this includes knowledge of visual descriptors. We thus can simply ask an LLM, much like a 5-year old asking their parent: what does it look like?

We provide an alternative to the current zero-shot classification paradigm with vision-language models, comparing to class descriptors obtained from a large language model instead of just the class directly. This requires no additional training, and does not require substantial computational overhead during inference. By construction, this provides some level of inherent interpretability; we can know an image was labeled a tiger because the model saw its stripes rather than its tail. Rather than compromising performance metrics, our approach improves accuracy across datasets and distribution shifts, achieving a $\sim$ 4-5% increase on top-1 ImageNet accuracy.

## 2 METHOD

### 2.1 PERFORMING CLASSIFICATION WITH DESCRIPTORS

Given an image $x$, our goal is to classify whether a visual category $c$ is present in the image, where we represent a category $c$ through a textual phrase, e.g., "school bus." To make our model both

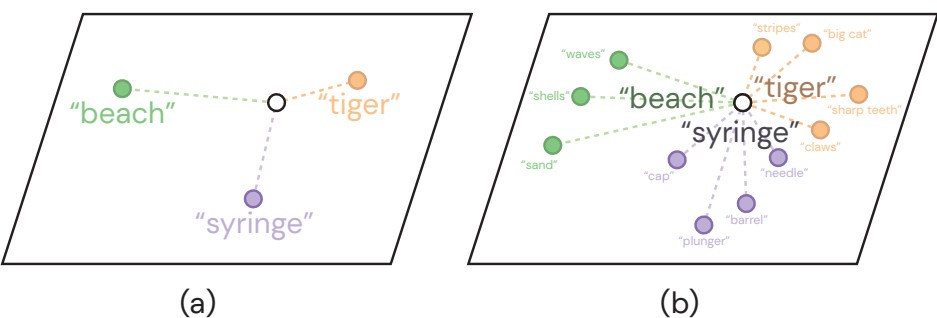

Figure 2: (a) The standard vision-and-language model compares image embeddings (white dot) to word embeddings of the category name (colorful dots) in order to perform classification. (b) We instead mine large language models to automatically build descriptors, and perform recognition by comparing to the category descriptors.

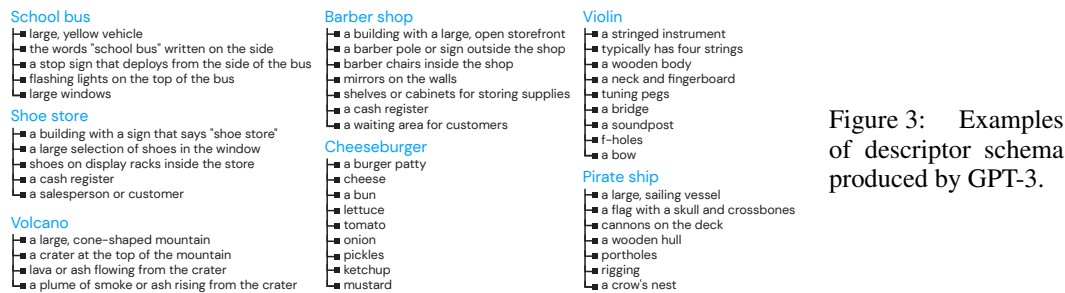

Figure 3: Examples of descriptor schema produced by GPT-3.

interpretable and editable, we estimate a score for category $c$ through the additive decomposition:

$$s(c, x) = \frac{1}{|D(c)|} \sum_{d \in D(c)} \phi(d, x) \tag{1}$$

where $D(c)$ is the set of descriptors for the category $c$ and $\phi(d, x)$ is the log probability that descriptor $d$ pertains to the image $x$. Our approach will represent the descriptors $d$ also through a natural language sentence; we explain how to obtain these in the next section.

This model $s(c, x)$ will output a high score when the dictionary for the category $D(c)$ contains many descriptors that highly match the observed image $x$. Figure 2 illustrates this approach to classification. We use addition so that some descriptors can be missing in the image, and normalize by the number of descriptors for the class to allow different classes to have different numbers of descriptors. Since the descriptors are both additive and expressed in natural language, the model is naturally interpretable. To understand why the model predicts category $c$, one can simply read which descriptors have a high score.

## 2.2 BUILDING DESCRIPTORS

For the classifier to work, we need to successfully estimate the descriptors $D(c)$ of each visual category. We propose to automatically construct this set by prompting a large language model, such as GPT-3, to describe the visual features that distinguish that object category in a photograph. We prompt the language model with the input:

```
Q: What are useful features for distinguishing a {category
   name} in a photo?
A: There are several useful visual features to tell there is a
   {category name} in a photo:
-
```

where `{category name}` is substituted for a given $c$. The generated list then comprises the dictionary $D(c)$. Further implementation details can be found in Appendix A.

Fig. 3 shows several example descriptor schemata that emerge from generative language pre-training. The descriptors often cover colors, shapes, object parts, counts, and relationships, but can be anything in natural language. While descriptors are closely related to more traditional "attributes," this flexibility distinguishes them, enabling each category's descriptors to be rich and nuanced. As we observe in Fig. 3, they can be category-specific, such as the "stop sign" for the "school bus" category, or more general, such as "cash register" for both "shoe store" and "barber shop."

While language models do not have images in their training set, they learn to imitate visual description successfully without visual input. The corpora used to train language models contain descriptions written by people with visual knowledge. These descriptions, aggregated at scale, provide a strong basis for visual recognition.

## 2.3 GROUNDING DESCRIPTORS

We use vision-language models to visually ground the natural language descriptors generated by the large language models, i.e., CLIP similarity to form $\phi$. Since descriptors are often relative to their class, we condition descriptors on the class name. For example, a "long tail" for a mouse

| Architecture for $\phi$ | ImageNet | | | ImageNetV2 | | | CUB | | | EuroSAT | | |
|---|---|---|---|---|---|---|---|---|---|---|---|---|
| | Ours | CLIP | $\Delta$ | Ours | CLIP | $\Delta$ | Ours | CLIP | $\Delta$ | Ours | CLIP | $\Delta$ |
| ViT-B/32 | **62.97** | 58.46 | 4.51 | **55.52** | 51.90 | 3.62 | **52.57** | 51.95 | 0.62 | **48.94** | 43.84 | 5.10 |
| ViT-B/16 | **68.03** | 64.05 | 3.98 | **61.54** | 57.88 | 3.66 | **57.75** | 56.35 | 1.40 | **48.82** | 43.36 | 5.46 |
| ViT-L/14 | **75.00** | 71.58 | 3.42 | **69.3** | 65.33 | 3.97 | **63.46** | 63.08 | 0.38 | **48.66** | 41.48 | 7.18 |
| ViT-L/14@336px | **76.16** | 72.97 | 3.19 | **70.32** | 66.58 | 3.74 | **65.257** | 63.41 | 1.847 | **48.74** | 44.80 | 3.94 |
| | Places365 | | | Food101 | | | Oxford Pets | | | Describable Textures | | |
| ViT-B/32 | **39.90** | 37.37 | 2.52 | **83.63** | 79.31 | 4.32 | **83.46** | 79.94 | 3.52 | **44.26** | 41.38 | 2.87 |
| ViT-B/16 | **40.34** | 38.27 | 2.07 | **88.50** | 85.61 | 2.90 | **86.92** | 81.88 | 5.04 | **45.59** | 43.72 | 1.86 |
| ViT-L/14 | **40.55** | 39.00 | 1.55 | **92.44** | 91.79 | 0.65 | **92.23** | 88.25 | 3.98 | **54.36** | 51.33 | 3.03 |
| ViT-L/14@336px | **41.18** | 39.58 | 1.59 | **93.26** | 92.23 | 1.03 | **91.69** | 88.20 | 3.49 | **54.95** | 52.39 | 2.55 |

Table 1: Accuracy gains over CLIP category name embedding baseline. We see a consistent $\sim$ 3-5% improvement across model sizes for ImageNet and ImageNetV2, as well as up to $\sim$ 7% on other datasets from dramatically different domains.

will be still shorter than a "short tail" for an elephant. We estimate similarity with text of the form `{category_name} which (is/has/etc) {descriptor}`. These text embeddings are similar to class prototypes (Snell et al., 2017; Rudin et al., 2021), but instead they are obtained across modalities, as discussed further in the Related Work. If an image belongs to the class, but does not show a particular descriptor, that descriptor is activated less. We show in Section 3.2 that we can nonetheless recognize new categories from descriptors that do not need such reference.

## 2.4 CLASSIFICATION AND EXPLANATION

Our model is able to discriminate between categories by selecting the one with the highest score:

$$\arg\max_{c \in C} s(c, x) \tag{2}$$

where $C$ is the set of object categories in the dataset. Since the model is required to construct predictions by first estimating descriptor similarities $\phi$, the model is explainable by construction. We can understand why a model picked one category by reading the descriptors that are activated in an image. The same mechanism allows us to also understand why a category was *not* selected.

## 3 EXPERIMENTS

Our experiments explore visual models with language as an internal representation. We show results for explainable object recognition, adaption to novel categories, and the reprogrammability of visual classifiers in order to correct biases and other errors. We quantitatively and qualitatively compare our method against CLIP, which is one of the most established methods for vision-language pretraining. To analyze the capabilities of our approach on real images, we consider a variety of domains, including everyday objects and satellite images.

## 3.1 EXPLAINABLE OBJECT RECOGNITION

We evaluate our method at the ability to perform image classification while also providing explanations for its decisions. While most interpretability methods come with a compromise on the benchmark performance, we demonstrate in Table 1 that our approach improves on it. Compared to CLIP which compares images to embeddings of class names, our approach improves performance by over 3% on average for ImageNet, without training on it.Furthermore, the improvements across a range of domains show that the advantages of our method is not limited to everyday object recognition. For example, we achieve up to $\sim$ 7% improvement on the EuroSAT dataset for satellite image recognition; a $\sim$ 2.5% improvement on the Describable Textures dataset for texture recognition; and a $\sim$ 1-2% improvement on the CUB benchmark for fine-grained classification of birds. This suggests that GPT-3 can provide some useful knowledge even for niche domains.

Since there is an inherent bottleneck to first construct linguistic attributes of an image, the method is naturally explainable. Fig. 1 shows several cases where the model justifies its explanation for the

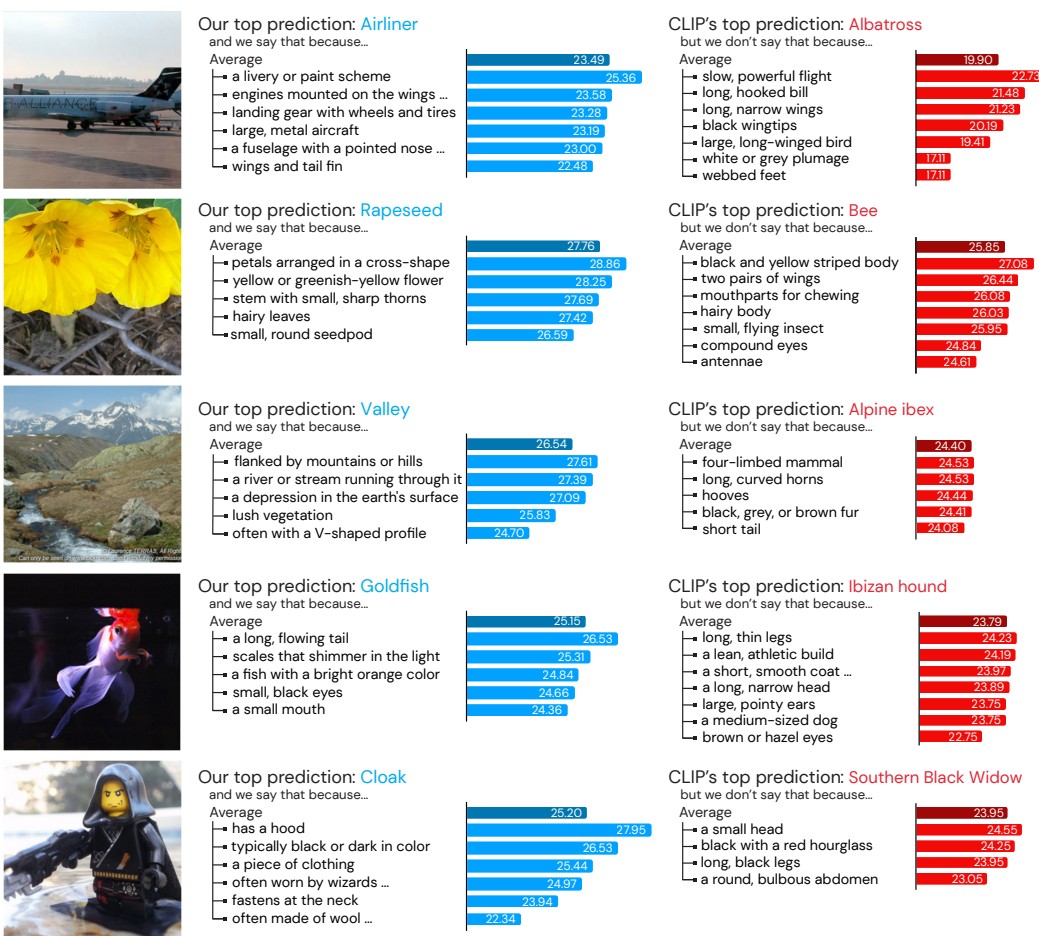

Figure 4: (left, in blue) We show example decisions and their justifications from our model. (right, in red) We show the prediction from CLIP, and the justification from *our model* why it did not select that answer. The bar charts show the descriptor similarity $\phi$ to the image in the CLIP latent space.

prediction. For example, in the top row, CLIP incorrectly classifies the airplane as an albatross, a choice our model disregards because it cannot identify features similar to a bird. Instead, our model correctly identifies there is an airliner because it can identify many features related to airplanes.

## 3.2 ACQUIRING AND UTILIZING NOVEL INFORMATION

A well-known limitation of machine learning models is that they often perform poorly on data they have not seen during training (Gulrajani & Lopez-Paz, 2020; Jiang et al., 2020; Wortsman et al., 2022; Djolonga et al., 2021). Although foundation models are trained on large portions of the Internet representing a wide variety of data, it is impossible for them to have been trained on concepts that only came into existence after they were trained.

In contrast, our approach acquires visual descriptions from a large language model, which allows it to build new classifiers for categories that $\phi$ has not encountered yet. CLIP was originally trained in February 2021, and we added two new categories to the ImageNet validation set that widely appeared on the Internet after this date: a) the Ever Given, which is the ship that blocked the Suez Canal in March 2021 (Wikipedia, 2022a), and b) the game Wordle, an online word game that went viral in January 2022 (Wikipedia, 2022c). For each category, we added five images into the existing validation set of $50,000$ ImageNet images.

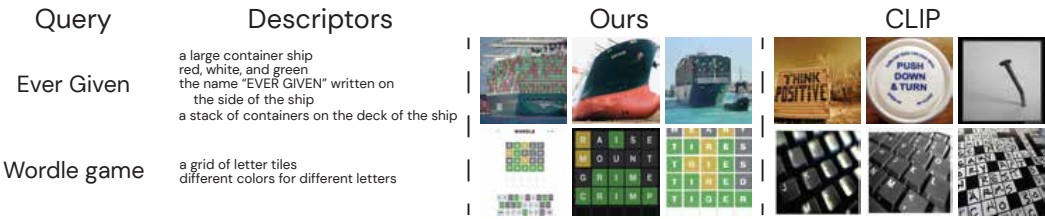

Figure 5: Our retrieved images vs those found by CLIP for concepts popularized after CLIP's training. If CLIP has not observed a particular text-image association when it was trained, it is bound to fail when retrieving images related to that text naïvely. Our approach queries the large language model to obtain the necessary external information. Incorporating this knowledge provides it cues for what to look for, allowing for successful retrieval of relevant images.

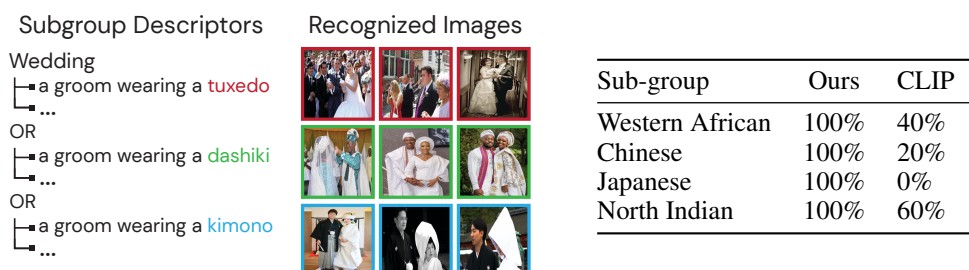

| Sub-group | Ours | CLIP |
|---|---|---|
| Western African | 100% | 40% |
| Chinese | 100% | 20% |
| Japanese | 100% | 0% |
| North Indian | 100% | 60% |

Figure 6: (left) CLIP only compares to the word 'wedding', yielding biased results – it only correctly recognizes the first row. The descriptor-based approach provides a way to address the bias, by expanding the initial set of descriptors (only the top) to be more inclusive with prior knowledge. (right) Modifying the descriptors to be more inclusive causes accuracy to significant improve on sub-groups.

We quantify the recall at retrieving these new examples within the top ten detections of the category, and Fig. 5 compares the performance of our system versus CLIP. For both categories, our method obtains 100% recall, while CLIP obtains 0% recall for Wordle and 10% recall for the Ever Given. Even though these categories are relatively new, GPT3 is able to build descriptors for them because enough people on the Internet have visually described them. By combining these descriptors together, the model can recognize the new category. In contrast, since these categories are novel to CLIP, baseline methods obtain poor performance. CLIP matches the "Wordle game" incorrectly category to keyboards instead of the game, likely an artifact of how CLIP tokenizes the input words. For the "Ever Given" category, CLIP retrieved just one correct example in the top ten detections. The example it retrieved correctly has its name written on the ship, and CLIP likely performed well on this example due its reading capabilities (Goh et al., 2021). However, for the rest of the nine Ever-Given ships in the dataset, CLIP was unable to correctly detect them with high confidence. This experiment illustrates that visual classifiers can be automatically built by leveraging the descriptive capabilities of large language models.

### 3.3 CORRECTING FAILURES BY DESCRIPTOR EDITING

Bias remains an unsolved challenge in machine learning, including for foundation models trained on large-scale data. Typically, diagnosing the source of biases is challenging because representations are usually black-box. However, linguistic attributes makes it possible to both identify which part of the system introduces a bias, and manually correct them in some cases.

We found that both CLIP and our model is biased towards Western celebrations for the "wedding" category. Inspired by the Inclusive Images Challenge (Atwood et al., 2020), we collected a challenge set consisting of images of weddings from various cultures to explore this bias. There are many wedding and cultural traditions in the world, and each exhibit their own identifying visual characteristics. On this dataset, CLIP tended to have a Western-centric bias, primarily considering

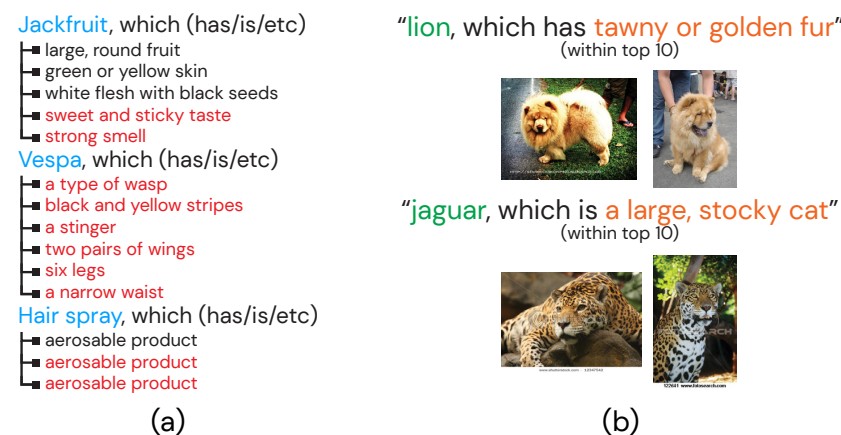

Figure 7: (a) Examples of errors in the descriptors produced by GPT-3. (b) Examples of undesired behavior in CLIP retrievals from descriptors. The text in red annotates the error.

"white weddings" (named for the white dress brides wear in this tradition) (Wikipedia, 2022b). Our model showed a similar bias. When our classifier prompted GPT3 for the visual descriptors of a wedding, the language model mostly prescribed that the groom should be wearing a tuxedo. For these models to be deployed and trusted, we must have ways to amend them to be inclusive.

Since the decisions from our model are based on human-readable text, changing them changes the decision process. Figure 6 illustrates that we can intervene to partially fix the bias. We can manually overwrite the attribute for "a groom wearing a tuxedo" to include other clothing traditions, such as a dashiki (traditional clothing in many parts of Western Africa) or a kimono (traditional clothing in many parts of Japan). This effectively points the VLM to what it should look for instead of relying solely on its biased association with the word wedding. We can thus build a more inclusive classifier by considering the descriptor similarity of every such sets of descriptors, then choosing the category "wedding" if any of them have the highest total descriptor similarity compared to background classes.

On this dataset and in these settings, the table in Fig. 6 shows that intervention can effectively correct this bias. Compared to CLIP, our approach obtains significantly higher accuracy at recognizing the cultural variation of wedding photographs. Figure 6 shows several qualitative examples where the model is reprogrammed with different clothing attributes.

### 3.4 UNDERSTANDING FAILURE MODES

We explore a few limitations of the method here. We discuss failures both in the language model $D(c)$ and failures in the grounding $\phi$.

**Failure in descriptor creation.** Typically, GPT-3 produces high-quality descriptors; however, we noted three types of failures, shown in Figure 7 (a). The first failure mode is a shortcoming in understanding modalities. Despite being asked for visual features of the category, GPT-3 will occasionally produce descriptors that, while correct, reflect other modalities. In Figure 7 (a), we observe "jackfruit" is given descriptors pertaining to taste and smell in addition to vision. As these cannot be seen, they are not useful cues for the vision-language model.

Another failure case is caused by word ambiguity. Many words have multiple meanings. GPT-3 produces descriptors solely from the name of the category; it cannot see any of the images in the dataset to tell which meaning of a word is intended. It picks one meaning to describe. This often succeeds, as it tends to pick the most common meaning – for example, the descriptors for "St. Bernard" relate to the dog breed instead of the religious figure, matching the visual category. In some cases, however, it can lead to descriptors that are completely unrelated to the visual category. The "Vespa" in Figure 7 (a) is an illustrative example of this. In ImageNet, this category refers to scooters made by the Italian designer brand Vespa. GPT-3 instead describes a wasp. "Vespa" is the Italian word for "wasp" – the multilingual capabilities of GPT-3 actually lead to an undesired result.

This type of mistake is to some extent unavoidable solely from text without further information about the category; using the name "Vespa scooter" does not lead to this error.

Finally, GPT-3 still (albeit rarely) produces syntactic errors such as verbatim repetition of the same descriptor (see "Hair spray" in Fig. 7 (a)). As large language models progress, we expect this failure mode to continue to diminish. Our framework readily adapts to such advances; it will only continue to work better as language models improve further.

**Failure in recognizing descriptors.** While for most descriptors, CLIP successfully matches the descriptor text to images, we find it can also retrieve unexpected images in certain circumstances. We identify two interesting types of failure in recognition, illustrated in Figure 7 (b). If a descriptor matches multiple categories, there are some cases where it can be strongly activated for categories outside the intended one even when the intended category name is included in the text to be embedded. For example, the descriptor "lion, which has tawny or golden fur" retrieves two Chow Chows in the top 10 activations. We hypothesize this is due to the confluence of two factors. The first is of course that it is likely Chow Chows are strongly associated with having "tawny or golden fur." But typically, this is not enough on its own, judging by other descriptors (see Appendix for examples). We believe the other relevant factor is having category names that are themselves related. The Chinese name for Chow Chows includes the word 'lion' in it; similarly to the "Vespa" case for GPT-3, there error may have a multilingual nature. (We note that the other descriptors for "lion" do not fit Chow Chows the way "golden fur" does, and do not retrieve images of the dogs).

Another failure mode we observe stems from both word ambiguity and the ability of CLIP to solve multiple tasks. In particular, just as its ability to read can impede its ability to recognize objects (Goh et al., 2021), we find this same ability can also impede recognizing descriptors. The second example, "jaguar, which is a large, stocky cat" retrieves jaguars correctly, but is strongly activated by *stock photos* of jaguars over jaguars which are themselves *stocky cats*. This is especially true when they contain the word "stock", but also occurs for other stock photos, due to the ambiguity between the word "stock" in how it applies to images rather than cats.

# 4 RELATED WORK

Vision-language models have grown to be a dominant paradigm for visual recognition with the release of CLIP (Radford et al., 2021), showing strong zero-shot performance on a range of benchmarks across distribution shifts. Other work such as ALIGN (Jia et al., 2021), FLAVA (Singh et al., 2022), Florence (Yuan et al., 2021), and more have since furthered this paradigm. The hallmark of these recent models is that they are trained on large-scale datasets of image-text pairs collected from the Internet. They have seen success on a variety of tasks, including classification, detection (Kamath et al., 2021), and more. Compared to previous models, vision-language models boast the advantage of connecting visual data to free-form language rather than fixed categories. Concurrent work Menon et al. (2022) shows that the visual representations learned by such models can be predisposed towards certain tasks a priori.

Interpretability and explainability for deep models in vision is a field too broad to fully cover here; we discuss some relevant examples here, and direct the interested reader to (Gilpin et al., 2019) for a broader overview. Much of the work in explaining deep model decisions is post-hoc, often in the form of heatmaps such as GradCAM (Selvaraju et al., 2020). Chefer et al. (2021) extend similar techniques to vision transformers, such as those often used for vision-language models. Such work can be useful for understanding low-level decision factors such as "which part of the input image was the decision based on?" However, as the decision is not constructed from these explanations but rather the explanations are constructed from the decision, there can be questions of *faithfulness*, i.e. how much the generated explanation actually reflects the decision process of the model. In addition, outputs such as heatmaps require some interpretation on the part of the user to parse, and cannot easily capture medium-to-high level factors such as "cash register" being the deciding factor for a 'store'. Other work aims to produce explanations in natural language for vision. Park et al. (2018) create multimodal explanations by training a language model to produce explanations in tandem with a visual classifier, but these text explanations need not be what the visual classifier bases its decision on, rather they intend to be plausible explanations for the given image. Sammani et al. (2022) extends this idea to more closely integrate the representations of both modalities.

Our approach is natural when viewed through the lens of prototype learning with neural networks (Vinyals et al., 2017; Snell et al., 2017; Chen et al., 2019), with some marked differences. In particular, our work is partly inspired by Chen et al. (2019), which constructs inherently interpretable decisions for visual classification by comparison to prototypical examples in the training set. This work bridges the gap between fully black box neural networks and fully transparent white box models without compromising performance; similarity to the prototypes is estimated with a black box neural network, but which prototypical images the decision was based on and how much for each of them is exposed to the user. This is similar to our use of descriptors at inference time, but requires a unique training procedure and produces only visual explanations. We can view $D(c)$ as a support set of prototype vectors for class $c$. Rather than prototypes in the typical sense, however, these are *text* prototypes for *visual* data. Work such as Chen et al. (2019) has previously explored prototypes for inherently interpretable models, but these have been "visual words" rather than free text. In addition, we do not need to learn our prototypes, instead making use of pre-trained foundation models. We note that one difference from ProtoNets (Snell et al., 2017) is that we compute the class score $s$ as the mean similarity to support vectors (i.e., descriptor) rather than the similarity to the mean of the support vectors. Aggregating similarities enables interpretability, as the class score can be decomposed into similarity with each descriptor, more akin to Chen et al. (2019); Vinyals et al. (2017). Comparing to the mean of support vectors is essentially the approach of typical "prompt ensembling." CLIP (Radford et al., 2021) presents a remarkable ensemble of 80 prompts hand-designed for the ImageNet dataset over the course of a year. We encourage readers to see concurrent work (Pratt et al., 2022) that demonstrates that prompt ensembling with prompts obtained by large language models can improve accuracy on recognition tasks. One of the key differences between Pratt et al. (2022) and our work is that while prompt ensembling is an effective tool for increasing accuracy, it does not afford the same interpretability, editability, or adaptability on a per-descriptor basis that our approach does.

There are several advantages to using text prototypes. Critically, using natural language allows us to obtain descriptors by leveraging the world knowledge condensed into large language models such as GPT-3. This eschews costly learning processes and incorporates external knowledge effectively. Text prototypes also are readily interpretable, whether by a technical user or a layperson. It is easier and more natural for a person to edit text than to edit visual data to a desired prototype, including to define a previously-unseen category.

Other work demonstrates that external text knowledge can provide substantial aid to vision tasks. K-LITE (Shen et al., 2022) shows that such knowledge from WordNet and Wiktionary has the potential to enhance prompts for vision-language models. (Zeng et al., 2022) is similarly motivated in connecting large language models to vision-language models, enabling emergent capabilities such as image captioning. Large language models, especially GPT-3 (Brown et al., 2020), have seen widespread application since their introduction due to their impressive ability to generate sequences similar to those observed from humans. PICA (Yang et al., 2021) demonstrates knowledge derived from GPT-3 can aid few-shot VQA tasks.

Our work is also closely related to zero-shot attribute-based classification, such as (Lampert et al., 2014). Parikh & Grauman (2011) develop a system to create a vocabulary of "nameable" attributes from humans. Romera-Paredes & Torr (2017) demonstrate a framework where connections between attributes and classes are given by the environment. Socher et al. (2013) introduce the idea of using word embeddings to use knowledge distilled from large-scale text corpora for zero-shot visual recognition; we go a step further and use large language models to obtain the words to embed themselves. These selected works share similar motivation to our use of large language models to create dictionaries of descriptors. As this area is also too large to summarize shortly, we direct the interested reader to Xian et al. (2020) for further reading.

## 5 CONCLUSION

We introduce a new framework for zero-shot classification with vision-language models. We leverage the linguistic knowledge about visual categories from large language models to generate textual descriptors for each category, comparing images to these descriptors rather than estimating the similarity of images directly with category names. Using GPT-3 and CLIP, we show promising results showing the capabilities of this framework to provide interpretable model decisions, improve performance on recognition tasks, enable adaptation to new knowledge, and mitigate bias.

## 6 ETHICS

Large pretrained models such as GPT-3 and CLIP learn various biases relating to race, culture, gender, and more from the Internet Goh et al. (2021). Systems that use these models can reproduce or even exacerbate this bias. Using both in tandem has the potential to compound the biases of both models. Interpretable models, like the one we present, have the potential to shed light on these biases that could otherwise remain unknown. For example, it is likely that descriptor dictionaries produced by GPT-3 could reflect its biases. We hope that the methods we present for editability and bias mitigation serve as useful tools to combat said biases.

## 7 REPRODUCIBILITY

We use CLIP as our vision-language model and GPT-3 as our large language model, both of which can be queried by anyone – CLIP having weights available, and GPT-3 having a public API. Sufficient details to reproduce the method can be found in Section 2 (for inference) as well as Appendices A (for querying language models) and B (for editability). We will release the data for the editability and adaptability experiments where the appropriate licenses permit and sufficient de-identification is possible. We will also release code upon publication.

## 8 ACKNOWLEDGEMENTS

This research is based on work partially supported by the NSF NRI Award #2132519, the DARPA MCS program, and the DARPA GAILA program. SM is supported by the NSF Graduate Research Fellowship. We'd like to thank Dídac Surís and Rich Zemel for helpful discussions and feedback.

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

## A  PROMPTING THE LANGUAGE MODEL

A key aspect of our method is producing discrete, separate descriptors through the language model, rather than embedding everything together as is typical for prompting. Thus, an important consideration is how to encourage the language model to produce descriptors in a way that does not require human parsing. Recall our prompt structure is of the form

```
Q: What are useful features for distinguishing a {category name} in a
    photo?
A: There are several useful visual features to tell there is a {category
    name} in a photo:
-
```

We find that adding the trailing '-' is enough to typically result in a bulleted list output. This is simple to automatically obtain descriptors from by simply removing the hyphens.

Per OpenAI's API recommendations, we use the structure `Q:` `A:` for the query and the desired response. We sample from the 'text-davinci-002' model with temperature of $0.7$ and a maximum token length of $100$.

As with previous work with GPT-3, we find that the list formatting becomes more reliable when one or two examples of desired output are provided. These can be of the form

```
Q: What are useful visual features for distinguishing a lemur in a photo?
A: There are several useful visual features to tell there is a lemur in a
    photo:
- four-limbed primate
- black, grey, white, brown, or red-brown
- wet and hairless nose with curved nostrils
- long tail
- large eyes
- furry bodies
- clawed hands and feet
```

which itself was constructed by GPT-3 (although such examples can easily be constructed by hand as well).

Presumably, all of this could be improved with more effort towards prompting; we did not tune these prompts at all after our initial generation of descriptors for the 1000 ImageNet classes.

## B  DATASET DETAILS

We consider the ImageNet dataset (Russakovsky et al., 2015) for everyday object recognition; ImageNetV2 (Kornblith et al., 2019) for distribution shift from ImageNet; CUB for fine-grained classification of birds (Wah et al., 2011); EuroSAT (Helber et al., 2019) for satellite image recognition; Places365 for scenes; Food101 (Bossard et al., 2014) for food; Oxford Pets (Parkhi et al., 2012) for common animals; and Describable Textures Cimpoi et al. (2014) for in-the-wild patterns.

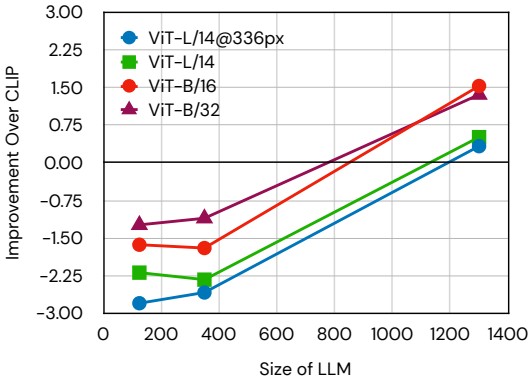

Figure 8: We use the OPT Zhang et al. (2022) series of language models to evaluate the influence of different language models at different parameter counts. We find that using descriptors from smaller language models can actually hurt performance, while after a certain size performance gains appear; this trend holds across sizes of VLM. We hypothesize this is because smaller models do not contain the knowledge of the visual world that larger language models possess.

## C    FURTHER COMPARISON

In this section, we provide some additional interesting quantitative comparisons.

| | ImageNet | | | |
|---|---|---|---|---|
| | Wiktionary Descriptions | Wordnet Descriptions | Ours | CLIP |
| ViT-B/32 | 57.91 | 60.00 | **62.97** | 58.99 |
| ViT-B/16 | 62.56 | 64.58 | **68.03** | 64.05 |
| ViT-L/14 | 68.87 | 71.14 | **75.00** | 71.57 |
| ViT-L/14@336px | 69.21 | 72.16 | **76.16** | 72.96 |

Table 2: Comparison with auxiliary information obtained from WordNet and Wiktionary rather than GPT-3. We observe that the Wiktionary information tends to hurt performance, while WordNet sometimes slightly helps and sometimes slightly hurts. We hypothesize this is because the information contained in WordNet and Wiktionary concerns definitions more often than visual descriptions.

## D    EDITABILITY/BIAS MITIGATION EXPERIMENT DETAILS

We collect 10 images depicting each of four wedding traditions from Western Africa, China, Japan, and Northern India. We use Flickr to collect these images with Creative Commons licensing, with the exception of the Western African examples, which could not be found on Flickr; for these, we

| | Aggregation Method | |
|---|---|---|
| | Mean | Max |
| ViT-B/32 | **61.34** | 60.00 |
| ViT-B/16 | **66.45** | 64.58 |
| ViT-L/14 | **73.15** | 71.14 |
| ViT-L/14@336px | **74.19** | 72.16 |

Table 3: Comparison between aggregating descriptor similarities for a given category using their mean (as described throughout the paper) and their maximum. We find the mean to have a consistent benefit over the max, suggesting using multiple justifications in conjunction provides some benefit.

|              | ImageNet (80 Prompts) | | |
|--------------|--------|-------|------|
|              | Ours   | CLIP  | Δ    |
| ViT-B/32     | **63.76** | 63.37 | 0.39 |
| ViT-B/16     | **68.83** | 68.36 | 0.47 |
| ViT-L/14     | **75.96** | 75.52 | 0.44 |
| ViT-L/14@336px | **76.85** | 76.57 | 0.28 |

Table 4: Comparison with ImageNet using all 80 hand-engineered prompts created for the original CLIP paper. Though the prompts are not hand-tuned to descriptors, we note that they still provide some benefit; this motivates future work creating hand-tuned prompts explicitly designed for descriptor-based methods.

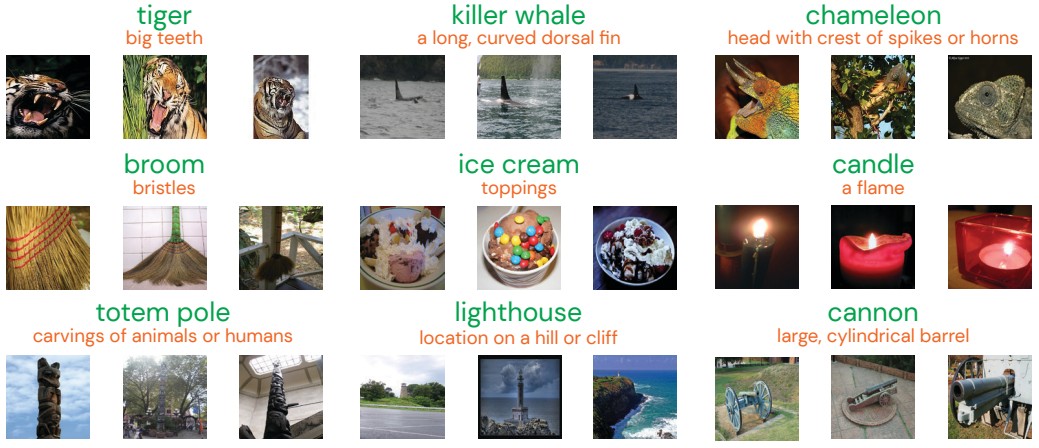

Figure 9: Top retrievals for various descriptors.

use Google Images. (We will release the images for which licenses are provided with identifying features blurred.)

To evaluate performance, we add the category 'wedding' to the 1000 ImageNet categories. We provide descriptors corresponding to each of the edited versions of the original, GPT-3-constructed descriptors of a white wedding. We perform these edits by identifying cross-cultural analogs in each descriptor and replacing the Western-specific words, such as 'tuxedo' → 'dashiki' for the Western African example. This results in 5 subgroups, including each of the four additional cultures and the original Western-centric descriptors; if the average descriptor score for any of these 5 is the highest, the category chosen is 'wedding.' We remove the existing category 'bridegroom' as this leads to category overlap, but the more general 'wedding' need not include only men. We use the CLIP RN50 model in these experiments, but find similar results across model sizes.

## E  TOP DESCRIPTOR ACTIVATIONS

Here we show various descriptors' top retrievals in Fig. 9.

