# OpenReview forum: "Visual Classification via Description from Large Language Models"
_ICLR.cc/2023/Conference — ICLR 2023 notable top 5%_

### Official Review · Reviewer_UNQW · 2022-10-13

**Confidence:** 5
**Correctness:** 3
**Technical Novelty And Significance:** 2
**Empirical Novelty And Significance:** 3
**Recommendation:** 8

**Clarity, Quality, Novelty And Reproducibility:**

- The proposed method is described clearly. Due to its simplicity, it should be easy to reproduce the reported result. The authors also promise to release code upon acceptance.
- The novelty of the propose method is limited. The proposed method is more like a "clever trick" rather than a fundamental technical innovation.

**Strength And Weaknesses:**

Strengths:
- The proposed idea is simple and well-executed. The paper is also well-written.
- Visual-prompt-engineering can be a heuristic and tedious process. It is refreshing to see that LLMs can help to automate this process. The prompt for GPT-3 is also well-designed to guide the model in generating visual descriptors.
- The method provides interesting additional capabilities such as explainability and novel category classification.

Weakness:
- Could the proposed prompt-generation method generalize to other domains other than common objects and animals? It is necessary to evaluate on more datasets as in the CLIP paper, such as food/scene/satellite/medical images. In order for the method to be practical, it should at least not be harmful on the more difficult domains.
- It should be made clearer (rather than in the footnote) that the CLIP results in Table 1 uses a single base prompt, which I assume is "a photo of {class_name}"? Does the proposed method include the same base prompt with class_name?
- It is suggested to include the result for the hand-crafted CLIP prompts in Table 1.
- There exist other methods to construct visual descriptors that can offer similar advantages such as explainability. For example, using WordNet (K-LITE) or wikipedia articles [a]. It would be good to see some comparison between descriptors generated by GPT-3 and descriptors retrieved from knowledge base.

[a] Large-Scale Zero-Shot Image Classification from Rich and Diverse Textual Descriptions. Bujwid and Sullivan. ACL 2021.

Minor suggestions:
- One of the entry in Table 1 has an inconsistent number of digits.



**Summary Of The Paper:**

This paper proposes a simple idea to improve zero-shot image classification using CLIP-like models. It uses a pre-trained LLM (GPT-3) to automatically generate text descriptors for visual categories. The text descriptors are then used by CLIP to compute image-text similarities as prediction scores for each category. The method shows improved zero-shot classification performance on ImageNet and CUB datasets over the naive CLIP prompt.

**Summary Of The Review:**

This paper opens up an interesting new direction that uses LLMs to generate visual descriptors for CLIP-like models. I have two major concerns:
- The ability of the proposed method to generalize to other domains.
- Comparison with CLIP's prompt is not entirely clear. Comparison with knowledge-based method could be better.

I will happily raise more score if my concerns can be cleared.

---

> ### Author Response · Authors · 2022-11-16
> **Response to Reviewer UNQW**
>
> Thank you for your helpful comments! We’re glad that you thought the “idea is simple and well-executed,” “refreshing,” and “provid[ing] interesting additional capabilities.”
>
> Many of your suggestions ended up improving the paper to the point we thought all the other reviewers should also see it, so they are addressed in the note to all reviewers; we add some details point by point here.
>
> – Could the proposed prompt-generation method generalize to other domains other than common objects and animals?
> Thanks to this comment, we have examined many more domains in the new revision; we find our method helps just as much (and sometimes even more) on these other suggested domains. Please see the comment to all reviewers for the details. We think this finding makes the paper much stronger and we appreciate that you led us to it.
>
> – Table 1 CLIP prompting
> Please see the comment to all reviewers.
>
> – Human-developed knowledge base descriptors
> We now compare to this for WordNet and Wiktionary as you suggested (details in the comment to all). We also include qualitative comparison of the descriptors. We find that for both cases, they provide a definition which papers like K-LITE leverage, but they do not provide a list of distinct descriptors the same way we obtain from GPT-3, making explainability difficult.
>
> We hope that given these improvements to the paper in response to your initial concerns, you would increase your score.

---

> > ### Comment · Reviewer_UNQW · 2022-11-16
> > **Thanks for improving the paper**
> >
> > The authors' response addresses all of my concerns. I appreciate the authors for performing additional experiments. I would happily raise my score given that the revised paper is stronger than the previous version.

---

> > > ### Author Response · Authors · 2022-11-16
> > > **Thanks**
> > >
> > > Thank you for your updated review!

---

### Official Review · Reviewer_4Ax6 · 2022-10-24

**Confidence:** 4
**Correctness:** 3
**Technical Novelty And Significance:** 3
**Empirical Novelty And Significance:** 2
**Recommendation:** 8

**Clarity, Quality, Novelty And Reproducibility:**

The paper is overall an easy and enjoyable read. As highlighted above there a few missing details which might make exact reproducilibilty hard, but hopefully the idea is resistant to the exact method of similarity etc.


The main novelty of the work is in using GPT-3 descriptions of a class for both defining it and then subsequently using them to explain the content of the image.

**Strength And Weaknesses:**

Strengths:

- The paper is generally well written and a fun read. It has a interesting experiments and shows that interesting results in explainability can be achieved almost for free from manual intervention by harnessing GPT-3 descriptions and models in the mould of CLIP.

- A nice set of interesting experiments showing that the CLIP results on ImageNet can be improved upon from text descriptions automatically obtained from GPT-3 as opposed to the manually specified in the original CLIP paper. There is just one reservation here in that it is not clear which text prompts were used for the CLIP results in table 1.

Weaknesses:

- The original CLIP paper has already demonstrated that prompt engineering can have a significant impact on the zero-shot recognition results. So this finding in the paper is not new.


- The experimental evidence presented is nice. But some of these experiments are a little sparse and a little artificial. This is especially true for the "acquiring and utilizing novel information" learning experiments. It is fun to see that "Wordle" and "Ever Given" images can be recognised from their GPT-3 descriptions but it is obviously far from a complete study of the model's capabilities on this front and it is a bit unclear what it demonstrates. "Ever Given" is a proper noun and the name of a specific container ship. It is not surprising that CLIP cannot do image retrieval based on a proper noun which was generally unknown at the time of training CLIP.

   * The paper mentions that very little in the way of prompt engineering is done. The engineering that is performed is mainly done to ensure a list of descriptions is created by GPT-3. How sensitive are the results to the temperature and max token length used in the language model? Were the default values used or were they set using some form of hyper-parameter sweep.


- There are some technical details missing or brushed over. So for example:

   * In equation (1) how exactly is \phi(d, x) computed? Is a cos similarity used, does it use a softmax to compute the probabilities, etc...?

   * For table 1 for the CLIP column what is the text prompt used to acquire the classification? Is it just a classname or does it use the prompt "A photo of a {label}" as suggested in the CLIP paper as the basic prompt that boosts performance over just using the classname.

   * In figure 1 and figure 4 there are bar charts showing the compatibility between a text description and an image. What exactly are the numbers being displayed? Also the horizontal scaling seems to be different between the different examples and thus gives an inconsistent visual cue via the length of the bars.

- Figure 4 gives some qualitative results for the explainability for particular images. But it would be nice to have some more quantitative measures of how well the explainability is doing or thoughts on how to do this or show which explanations are most useful for each class etc... Also would some calibration be required to decide which descriptions are actually "valid/present" for a given image.

**Summary Of The Paper:**

This paper presents a simple but effective approach to zero-shot learning and explainability for image classification. The title of the paper says it all, but for completeness here is a summary. The main idea is to query a large language model, in particular GPT-3, to obtain multiple short descriptions about a class. These short descriptions are then projected into a joint embedding space which was learned to encourage visual grounding via multi-modal training pairs (ie CLIP). Then when an image from this class is encountered it is projected to the joint embedding space and its similarity to each class is computed by aggregating the similarity of the image's encoding to each encoding of the class's short descriptions. The image is then classified as the class that gives the highest overall similarity score. A by-product of this system is that the short descriptions with the highest scores then give a ready made interpretable explanation of why the system made its decision.

The contributions of the paper:

1) The paper demonstrates, and adds to the evidence, that the publicly available large language and multi-modal models, trained on large scale data such as CLIP, can be leveraged via descriptions (and prompt engineering) to perform explanability without a drop in classification performance and zero-shot learning. The interesting part is that the amount of manual prompt engineering can be minimized and can be performed by GPT-3 instead of by a human.

2) The paper raises interesting questions to think about how to best leverage the large scale foundational models in vision and language to perform zero-shot learning especially in a world with ambiguities and that is evolving constantly and how best to transfer knowledge from large scale language models to visual models.

**Summary Of The Review:**

My recommendation is based on the fact that I think the results achieved are interesting and stimulating for a reader with an interest in zero-shot learning. There are some weaknesses in the presentation of the technical and some of the experimental evidence seem more like tidbits as opposed to fully fleshed out ideas. However, I think overall I would be in favour of publication with some cleaning up of the presentation.

(
Typos spotted along the way

* Page 5 last line: valiation --> validation
* In section 4 on the 2nd line of the 2nd paragraph there is an empty reference.
)

Response to the authors' feedback:
I would like to thank the authors for their responses and the extra information and results provided. They have addresses my concerns I have now upgraded my rating accordingly. Well done on an interesting and fun paper!

---

> ### Author Response · Authors · 2022-11-16
> **Response to Reviewer 4Ax6**
>
> Thank you for your detailed, helpful feedback! We’re glad that you found it “well written and a fun read” as well as “interesting and stimulating for a reader with an interest in zero-shot learning.” We address your thoughts point by point below.
>
> – The original CLIP paper has already demonstrated that prompt engineering can have a significant impact on the zero-shot recognition results. So this finding in the paper is not new.
> You’re entirely right that prompt engineering’s impact on zero-shot recognition results is well-known! We’d like to clarify that we’re not claiming this broad observation as one of our contributions. Previous work does not incorporate descriptive information about the categories being considered, such as that tigers have stripes, instead using general words like “a tattoo of a {category}”. Our contribution is showing this auxiliary information can help, and providing a pipeline to obtain and use it with no human intervention.
>
> – It is not surprising that CLIP cannot do image retrieval based on a proper noun which was generally unknown at the time of training CLIP.
> We agree it’s not surprising that CLIP fails at this – that’s exactly why our method is needed. The interesting part of the Wordle/Ever Given experiments is that you can successfully get CLIP to recognize previously-unseen categories from the GPT-produced descriptors alone. We hope that future work creates more comprehensive datasets of concepts unseen by big models, but as of right now we are not aware of any such work, and there are not many concepts CLIP hasn’t seen. As time goes on, more novel concepts will emerge in the world, and our approach allows vision-language models to adapt with minimal overhead.
>
> – The paper mentions that very little in the way of prompt engineering is done. [...] How sensitive are the results to the temperature and max token length used in the language model? Were the default values used or were they set using some form of hyper-parameter sweep.
> We used the standard values for everything and it simply worked as-is, so we did not see a need to change them. No hyper-parameter sweep was necessary, but this could further improve the empirical performance. Since everything worked on the first try, the entire project only required ~$20 in GPT-3 credits.
>
> – In equation (1) how exactly is \phi(d, x) computed?
> It is the similarity in the CLIP latent space between the text embedding of descriptor $d$ and the image embedding of image $x$ (defined in the original CLIP work as the cosine similarity multiplied by 100). We’ve added a note on that in that section now.
>
> – Table 1 questions
> We use the base classname there, as the “a photo of a” prompt actually helps on ImageNet but hurts on some other datasets. We now show a comparison with all 80 prompts in the Appendix.
>
> – Figure 1,4 bar chart numbers; horizontal scaling
>
> The numbers are the CLIP similarity $\phi$ (defined in the original CLIP work as the cosine similarity multiplied by 100) between the image and the relevant descriptor, which provide an interpetable view of the model’s decision process. The horizontal scaling differs as the relative magnitude can only be compared for a given image rather than across images.
>
> – Quantitative measures
> Please see the comment to all reviewers for our improved quantitative evaluation. We agree that current measures of explainability are lacking in the era of vision-language models, and leave developing new, more suitable measures to future work.
>
> – Typos
> Thanks for catching these! We’ve corrected them now.
>
>
> Given these changes, we hope you would increase your score.

---

### Official Review · Reviewer_nRmq · 2022-10-24

**Confidence:** 4
**Correctness:** 3
**Technical Novelty And Significance:** 3
**Empirical Novelty And Significance:** 3
**Recommendation:** 6

**Clarity, Quality, Novelty And Reproducibility:**

+ Clarity: The paper is in general well written and the given examples are illustrative. I am unsure how to interpret the values of predicted score though (See weaknesses).

+ Quality: The analysis is great. The experiments need more quantitative results (See weaknesses).

+ Novelty: The paper is novel. Besides improving the CLIP baseline on public benchmarks (ImageNet, ImageNet-V2 and CUB), it also presents new applications such as retrieving novel classes and mitigating biases by editing text description.

+ Reproducibility: The method is conceptually simple and the CLIP model is publicly available. But it would be great if the authors can share the list of text descriptions (since GPT-3 is not fully publicly available).

**Details Of Ethics Concerns:**

no.

**Strength And Weaknesses:**

### Pros:

+ It is a conceptually novel to obtain text descriptors for image recognition by querying large language models.

+ The classification via text description naturally provides explanations for decision of objection recognition (Sec 3.1) and can potentially reduce existing biases in existing methods (Sec 3.3).

### Cons:

- The performance from the original CLIP is a weak baseline which does not use prompt ensembles. To obtain the text descriptors from GPT-3, we still needs to prompt and it is also mentioned in Appendix A that "the list formatting becomes more reliable when one
or two examples of desired output are provided" which needs additional manual efforts. That's why we need to allow the CLIP baseline to use better-designed prompts as well.

- What's the unit of values in the bar plots (Fig 1, 4)? From Eq (1), if $\phi(d, x)$ refers to a log probability, then the score would be in the range of $(-\infty, 0]$, no?

- The quantitative experiment is weak. I would classify Sec 3.2-3.4 as qualitative analysis since they are not evaluated on well-recognized benchmarks. I think it would strengthen the paper if there is more quantitative experiment other than image classification. Maybe long-tail object detection on LVIS or so?

- I didn't fully get the statement "Our approach does not fit on the ImageNet training dataset"? Do you mean that your method is not trained on ImageNet train set (equivalent to a zero-shot setting)?

**Summary Of The Paper:**

The paper propose an image recognition paradigm named "classification by description". To be specifically, instead of matching an image with its categorical description, the method first queries a large language model to obtain some textual descriptions for each category and base the recognition decision on these descriptors through consensus. In addition to interpretability, the method boosts recognition accuracy on the ImageNet and ImageNetV2. The paradigm is also able to recognize novel concepts and mitigate bias by descriptor editing.

**Summary Of The Review:**

This is generally a well-written paper with a novel approach to achieve both interpretability and performance boost. It would be better if there is more quantitative experiment to support.

---

> ### Author Response · Authors · 2022-11-16
> **Response to Reviewer nRmq**
>
> Thank you for your attentive comments! We are glad you thought “the analysis is great,” and “the paper is novel.” We address your feedback point by point below.
>
> – Improving quantitative experiments and baselines
> Thanks to your comments about this. In the revision, we have included substantially more quantitative experiments and new baselines (please see the comment to all reviewers). One additional note to your specific feedback: you note “it is also mentioned in Appendix A that "the list formatting becomes more reliable when one or two examples of desired output are provided" which needs additional manual efforts.” We emphasize that these examples are actually made by GPT itself, not by human hand-design, so we actually require very minimal hand-tuning. We also note the same prompt for generating descriptors works across datasets, whereas hand-crafted prompts only work if designed specifically to each dataset. We’ve now clarified this better.
>
> – “What's the unit of values in the bar plots (Fig 1, 4)? From Eq (1), if $\phi(d,x)$ refers to a log probability, then the score would be in the range of $(-\infty,0]$, no?”
> Good catch! The values in the bar plots are the CLIP similarity in latent space (defined in the original CLIP work as the cosine similarity multiplied by 100) prior to softmaxing, as these numbers are more meaningful for interpretation. If you convert them to log-probabilities, you would be correct about the ranges. We clarified this in the captions in the revision.
>
> – “I didn't fully get the statement "Our approach does not fit on the ImageNet training dataset"? Do you mean that your method is not trained on ImageNet train set (equivalent to a zero-shot setting)?”
>
> You are correct, we just meant we do not train on ImageNet. We’ve clarified this sentence now.
>
> – “It would be great if the authors can share the list of text descriptions”
> We are happy to share the text descriptions with the code release.
>
> Given these changes and your summary comment that “It would be better if there is more quantitative experiment to support,” we hope this can convince you to increase your score.

---

### Official Review · Reviewer_LAah · 2022-10-25

**Confidence:** 3
**Correctness:** 3
**Technical Novelty And Significance:** 3
**Empirical Novelty And Significance:** 3
**Recommendation:** 8

**Clarity, Quality, Novelty And Reproducibility:**

Novelty: to my best knowledge, this is an interesting first exploration of using off-the-shelf models without any training to improve classification tasks. Even though there are a few questions unanswered, this paper overall suggests and interesting and important direction.
Quality: it shows clear improvements over the baseline, but I think this is method's performance could be further improved.
Reproducibility: (this is from the paper):
We use CLIP as our vision-language model and GPT-3 as our large language model, both of which can be queried by anyone – CLIP having weights available, and GPT-3 having a public API. Sufficient details to reproduce the method can be found in Section 2 (for inference) as well as Appendices A (for querying language models) and B (for editability). We will release the data for the editability and adaptability experiments where the appropriate licenses permit and sufficient de-identification is possible. We will also release code upon publication.


**Strength And Weaknesses:**

Strength:
1. This idea is quite novel and intuitive.
2. The experiments have clearly demonstrated the benefits of the proposed methods.

Weakness:
1. the aesthetics of the figures could be further improved.
some potential questions were not answered
1. generalization: how this generalizes to other tools other than CLIP and GPT?
2. how do you weight the prob. of a set of features linked with "or" logic? for example, in the appendix, the lemur could be "black, grey, white, brown, or red-brown", do you just sum these prob up? A better way might be using max?


**Summary Of The Paper:**

This paper explores using a list of category descriptions as a better representation for image categorization using VLMs such as CLIP. Decomposing the category word into a set of related descriptions could bring a better textural representation when doing matching. In addition, these descriptions could provide interpretability.  The paper also demonstrates more benefits such as improvements in accuracy on ImageNet across distribution shifts; recognizing unseen concepts, mitigating bias.

**Summary Of The Review:**

This paper proposes a simple and yet effective method to improve current clip image categorization quality.  The results are relatively preliminary but it proposes a novel direction and clearly demonstrated the benefits (interpretability, removing biases, etc.)

This paper might be stronger if it:
1. demonstrates if this method is generalizable to other big models
2. explore in more systematic treatment on how to combine descriptors in addition to simply adding them. (Eq.1)
3. provide more in-depth analysis on the potential different results on categories at different levels in a hierarchy  (e.g., dog vs husky)

---

> ### Author Response · Authors · 2022-11-16
> **Response to Reviewer LAah**
>
> Thank you for the positive feedback and useful suggestions! We are glad you find it to be an “interesting and important direction,” and noted the empirical benefits. We address your thoughts point by point below.
>
> – “The paper might be stronger if it demonstrates if this method is generalizable to other big models.”
>
> We thought this was an interesting point, and it also made us wonder how critical using ‘big’ models really is to our method. This led us to a new experiment considering alternative models and model sizes. Specifically, we use the OPT [citation] family of openly-available language models of various different sizes, and compare the performance using descriptors using different parameter-count models. We include the results in Appendix [APPENDIX]. We find even other models can lead to performance gains, but only above a certain size of model. Below this size, the descriptions can even hurt, which we can attribute to the smaller models’ lack of visual knowledge.
>
> – More systematic treatment on how to combine descriptors:
>
> We now compare three methods of combining descriptors. Originally, in the presence of multiple features, we would consider the mean score across descriptors, with the maximum across those linked with ‘or’ clauses. Your comment led us to consider an alternative: taking the maximum score across descriptors instead of the mean. We show the results in the Appendix. We find the mean to have a consistent benefit over the max, suggesting using multiple justifications in conjunction provides some benefit.
>
> – Hierarchy
>
> You make a good point that our approach naturally lends itself to hierarchical applications. We leave this as an interesting direction for future work.
>
> Given these encouraging new results, we hope you might be convinced to increase your score; we appreciate your feedback strengthening our paper!

---

### Author Response · Authors · 2022-11-16
**Comment to all reviewers**

We thank all reviewers for their thoughtful feedback, which we believe has led to a substantially stronger paper (with the revision now posted). We appreciate that reviewers found our work to be “novel and intuitive” (Reviewer LAah), “novel [...] [and] conceptually simple” (Reviewer nRmq), “fun” (Reviewer 4Ax6), and “simple and well executed” (Reviewer UNQW).

Multiple reviewers pointed out the paper might benefit from expanded quantitative evaluation. We agree with this point and so have conducted an additional suite of quantitative experiments.

We performed experiments on four additional domains (satellite imagery, textures, food, and pets). On the EuroSAT dataset for satellite imagery, we outperform the CLIP baseline by ~5%; on the Describable Textures Dataset, ~3%; on the Food101 dataset, ~4%; and the Oxford Pets dataset, ~3.5%. Note that we achieve these results with the prompt remaining unchanged – the same prompt used for ImageNet yields these improvements. This demonstrates that our approach can improve performance across a range of domains with no tuning, a stronger result than in our initial submission.

We also introduce two additional, enhanced baselines. Reviewer UNQW gave the interesting suggestion of comparing to descriptions retrieved from existing knowledge bases where possible. Thus, we now include comparison to “CLIP with WordNet Descriptions” and “CLIP with Wiktionary Descriptions.” (Due to space restrictions, these results can be found in the Appendix.) Comparison using WordNet knowledge  is only possible for ImageNet, where every category will have an entry in the knowledge base, whereas our use of GPT-3 allows for flexible generation of descriptions for arbitrary categories. Our approach outperforms both, showing a 4x relative improvement over them.

Some reviewers also asked about the hand-crafted prompts CLIP originally employs. We caution that these are extensively hand-tuned, overfit to ImageNet, and often opaque; for instance, one of the best hand-made prompts for ImageNet is “itap of {category}.” A list of all 80 prompts engineered for the original CLIP paper can be found at https://nbviewer.org/github/openai/CLIP/blob/main/notebooks/Prompt_Engineering_for_ImageNet.ipynb#Preparing-ImageNet-labels-and-prompts . These prompts also don’t transfer across domains. Our approach, in contrast, is simple and works across domains with NO prompt tuning. Nevertheless, we now include results using the 80 hand-crafted prompts in the Appendix; we still show improvement in this setting, and we are confident the gap can be widened substantially by also hand-crafting prompts for our approach, which we leave to future work.

---

> ### Author Response · Authors · 2022-11-16
> **Comment to all reviewers (2)**
>
> For comparison, we provide the original 80 CLIP hand-tuned prompts for ImageNet here, in comparison with our single prompt (for GPT-3) that succeeds across domains when employed with our method.
>
> Original CLIP:
> ```
>     'a bad photo of a {}.',
>     'a photo of many {}.',
>     'a sculpture of a {}.',
>     'a photo of the hard to see {}.',
>     'a low resolution photo of the {}.',
>     'a rendering of a {}.',
>     'graffiti of a {}.',
>     'a bad photo of the {}.',
>     'a cropped photo of the {}.',
>     'a tattoo of a {}.',
>     'the embroidered {}.',
>     'a photo of a hard to see {}.',
>     'a bright photo of a {}.',
>     'a photo of a clean {}.',
>     'a photo of a dirty {}.',
>     'a dark photo of the {}.',
>     'a drawing of a {}.',
>     'a photo of my {}.',
>     'the plastic {}.',
>     'a photo of the cool {}.',
>     'a close-up photo of a {}.',
>     'a black and white photo of the {}.',
>     'a painting of the {}.',
>     'a painting of a {}.',
>     'a pixelated photo of the {}.',
>     'a sculpture of the {}.',
>     'a bright photo of the {}.',
>     'a cropped photo of a {}.',
>     'a plastic {}.',
>     'a photo of the dirty {}.',
>     'a jpeg corrupted photo of a {}.',
>     'a blurry photo of the {}.',
>     'a photo of the {}.',
>     'a good photo of the {}.',
>     'a rendering of the {}.',
>     'a {} in a video game.',
>     'a photo of one {}.',
>     'a doodle of a {}.',
>     'a close-up photo of the {}.',
>     'a photo of a {}.',
>     'the origami {}.',
>     'the {} in a video game.',
>     'a sketch of a {}.',
>     'a doodle of the {}.',
>     'a origami {}.',
>     'a low resolution photo of a {}.',
>     'the toy {}.',
>     'a rendition of the {}.',
>     'a photo of the clean {}.',
>     'a photo of a large {}.',
>     'a rendition of a {}.',
>     'a photo of a nice {}.',
>     'a photo of a weird {}.',
>     'a blurry photo of a {}.',
>     'a cartoon {}.',
>     'art of a {}.',
>     'a sketch of the {}.',
>     'a embroidered {}.',
>     'a pixelated photo of a {}.',
>     'itap of the {}.',
>     'a jpeg corrupted photo of the {}.',
>     'a good photo of a {}.',
>     'a plushie {}.',
>     'a photo of the nice {}.',
>     'a photo of the small {}.',
>     'a photo of the weird {}.',
>     'the cartoon {}.',
>     'art of the {}.',
>     'a drawing of the {}.',
>     'a photo of the large {}.',
>     'a black and white photo of a {}.',
>     'the plushie {}.',
>     'a dark photo of a {}.',
>     'itap of a {}.',
>     'graffiti of the {}.',
>     'a toy {}.',
>     'itap of my {}.',
>     'a photo of a cool {}.',
>     'a photo of a small {}.',
>     'a tattoo of the {}.'
>  ```
>
> Ours:
>
> ```
> Q: What are useful features for distinguishing a {category name} in a photo?
> A: There are several useful visual features to tell there is a {category name} in a photo:
> -
> ```

---

### Decision · Program_Chairs · 2023-01-20

**Decision:**

Accept: notable-top-5%

**Justification For Why Not Higher Score:**

N/A

**Justification For Why Not Lower Score:**

I found the work to be very interesting. The idea is so simple, but intuitive and effective. It presents a new way to interact and understand systems, even for simple tasks like classification. An oral presentation at the conference would enable this idea and work to reach a broad audience. This paper can inspire future work in vision that can lead to similarly interpretable and effective models for more tasks.
The reviewers and scores were also very positive.

I am recommending an Oral, but I would also be in favor of a spotlight. I do feel that the idea and paper deserves more than a poster.

**Metareview: Summary, Strengths And Weaknesses:**

This paper presents Classification by Description, where a vision and language model for classification is asked to verify typical descriptions for object categories along with an overall category name. Using these descriptors results in a more robust model with improved ImageNet results and also a highly interpretable system.

Four reviewers provided reviews for this paper. Overall they were positive about this paper at the initial review. They enjoyed reading the paper, found the idea to be intuitive, simple and effective, found the idea of complementing image classification with description verification as novel and the presented experiments to be clear. Their main concerns revolved around the presentation of some of the material and requested additional experiments. The rebuttal addressed these points very well, which caused multiple reviewers to respond positively and raise their ratings. They are now in strong favor of this paper. Given the reviews, rebuttal, discussion, score and my own reading of the work to be interesting an novel, I am recommending an accept.

**Note From Pc:**

if the above contains the word "oral" or "spotlight" please see: "oral" presentation means -> notable-top-5% and "spotlight" means -> notable-top-25%. As stated in our emails, we are disassociating presentation type from AC recommendations

**Summary Of Ac-Reviewer Meeting:**

N/A